# Microbial Shift in the Enteric Bacteriome of Coral Reef Fish Following Climate-Driven Regime Shifts

**DOI:** 10.3390/microorganisms9081711

**Published:** 2021-08-11

**Authors:** Marie-Charlotte Cheutin, Sébastien Villéger, Christina C. Hicks, James P. W. Robinson, Nicholas A. J. Graham, Clémence Marconnet, Claudia Ximena Ortiz Restrepo, Yvan Bettarel, Thierry Bouvier, Jean-Christophe Auguet

**Affiliations:** 1UMR MARBEC, Université de Montpellier, CNRS, Ifremer, IRD, 34095 Montpellier, France; sebastien.villeger@cnrs.fr (S.V.); marconnet.clemence@gmail.com (C.M.); claudiaximenaro@gmail.com (C.X.O.R.); yvan.bettarel@ird.fr (Y.B.); thierry.bouvier@cnrs.fr (T.B.); Jean-christophe.AUGUET@cnrs.fr (J.-C.A.); 2Lancaster Environment Centre, Lancaster University, Lancaster LA1 4YQ, UK; christina.hicks@lancaster.ac.uk (C.C.H.); james.robinson@lancaster.ac.uk (J.P.W.R.); nick.graham@lancaster.ac.uk (N.A.J.G.)

**Keywords:** coral-macroalgal shift, coral reef fish, enteric bacteriome, microbial functions, barcoding

## Abstract

Replacement of coral by macroalgae in post-disturbance reefs, also called a “coral-macroalgal regime shift”, is increasing in response to climate-driven ocean warming. Such ecosystem change is known to impact planktonic and benthic reef microbial communities but few studies have examined the effect on animal microbiota. In order to understand the consequence of coral-macroalgal shifts on the coral reef fish enteric bacteriome, we used a metabarcoding approach to examine the gut bacteriomes of 99 individual fish representing 36 species collected on reefs of the Inner Seychelles islands that, following bleaching, had either recovered to coral domination, or shifted to macroalgae. While the coral-macroalgal shift did not influence the diversity, richness or variability of fish gut bacteriomes, we observed a significant effect on the composition (R2 = 0.02; *p* = 0.001), especially in herbivorous fishes (R2 = 0.07; *p* = 0.001). This change is accompanied by a significant increase in the proportion of fermentative bacteria (*Rikenella, Akkermensia*, *Desulfovibrio*, *Brachyspira*) and associated metabolisms (carbohydrates metabolism, DNA replication, and nitrogen metabolism) in relation to the strong turnover of *Scarinae* and *Siganidae* fishes. Predominance of fermentative metabolisms in fish found on macroalgal dominated reefs indicates that regime shifts not only affect the taxonomic composition of fish bacteriomes, but also have the potential to affect ecosystem functioning through microbial functions.

## 1. Introduction

Coral reefs have increasingly been subject to critical disturbances leading to a decrease of coral cover [1], a loss of coral habitat biodiversity [2], and to a reduction in associated ecosystem services [3,4]. Among the multiple stressors driving reef ecosystem decline, sea surface warming is responsible for severe bleaching events worldwide and the subsequent mortality of corals. In addition to climatic anomalies, overexploitation of herbivore fishes and nutrient discharges derived from land run-off can reduce coral cover and enhance the proliferation of macroalgae [5,6]. Indeed “coral-macroalgae regime shifts” are frequent in post-disturbance reefs [7,8]. Increase in macroalgal cover affects the resilience of coral reefs by reducing the survival and growth of adult corals [9], and/or preventing the recruitment of juvenile corals [10]. Macroalgae also produce secondary metabolites that can induce the growth of pathogenic and fouling microorganisms, causing a physiological deterioration of the coral tissues [11] and a dysbiosis in their microbiome [12]. This shift is not only dramatic for coral fitness, but it also impacts the assemblage composition and trophic structure of the entire coral habitat [13,14] and endangers associated ecosystem services (i.e., protection of coastal communities against storms, provision of protein through reef fisheries, and generation of tourism related incomes) [3,15,16].

Among coral reef biota, fishes play a well-known central role in coral-macroalgae regime shifts since the loss of herbivorous fishes through overfishing is considered as one of the causes of dominance by macroalgae [5]. Changes in composition and abundance of fish assemblages related to coral-macroalgae regime shifts are well understood, leaving gaps in knowledge about the impact of macroalgal dominance on other ecological traits of fishes, such as their microbiota. The great diversity of coral reef fishes, with more than 6000 species described [17], combined with the high diversity of their biological traits, provide specific ecological niches both on their skin and within their bodies, which ultimately promote the development of taxonomically and functionally original microbial lineages compared to the surrounding environment [18,19]. In a recent study, Chiarello et al. (2020) [19] showed with a conservative estimation that coral reef animal microbiota may account for up to 2.5% of Earth’s prokaryotic diversity, representing a hotspot of microbial diversity.

While our understanding of some components of fish microbiota such as viruses, archaea, and protists remain limited, their bacteriome has been more extensively studied in the recent years [18,19,20,21,22]. Most of these bacteria reside in the intestinal tract where they form complex communities and provide a range of essential functions linked to development, immunity, health, protection against pathogen invasion, and even influence behavior [23,24,25]. However, the most obvious and important role is the contribution of the fish bacteriome to the degradation and assimilation of large and complex molecules [20,26,27]. Evidence has accumulated that the gut bacteriome is not just a random set of microorganisms, but rather a highly variable community depending upon intrinsic fish factors such as diet or genetic background and extrinsic environmental conditions [18,21,22,28,29]. Nonetheless, our understanding of fish bacteriome variability is still scarce compared to terrestrial animals and is even more rare concerning coral reef fishes [30,31]. For example, environmental degradation or modification, such as urban sprawl and captivity, are known to have dramatic consequences on the enteric microbiome by altering the diet of wild animals, and thus impacts host fitness as observed in black howler [32,33] and other vertebrates [34,35,36]. Whether this impact also takes place in marine animals, and particularly the fish enteric microbiome, is poorly documented. The influence of coral bleaching induced regime shifts on coral reef fish bacteriomes remains unresolved. Furthermore, the loss of the most vulnerable fishes (i.e., corallivores) may induce an erosion of reef prokaryotic richness and their related functions [19]. Such events are clearly case studies to address fish gut microbiome responses and plasticity to environmental degradation. Moreover, understanding which bacterial lineages and associated functions are lost, and if there is compensation by other lineages, it is essential to better understand the consequences of bleaching-induced regime shifts on the functioning of coral reef ecosystems in general.

Coral reefs in Seychelles are located in the northern gyre of the western Indian Ocean (WIO), and are periodically subject to high marine heat waves with extreme SST associated with both El Niño and the Indian Ocean Dipole [37]. Severe bleaching and consequent regime shift events have occurred since the early 1980s [38] with two mass bleaching events in 1998 and 2016 that caused >90% and 70% of cover coral loss, respectively [37,39,40,41]. Following the 1998 coral bleaching event, Seychelles coral reefs underwent divergent trajectories, either recovering to a live coral condition or undergoing regime shifts to fleshy brown macroalgal dominance [6]. Here, we use these alternate Seychelles coral reef conditions to investigate the consequences of coral-macroalgal phase shift on the diversity and the structure of fish gut bacteriomes. First, we explored the diversity, richness, and composition associated with macroalgae and the enteric coral reef fish core bacteriomes. Second, we assessed the consequences of the regime shifts on the diversity, variability, and composition of the core bacteriomes at both taxonomic and functional levels.

## 2. Materials and Methods

### 2.1. Study Area and Sample Collection

Seven locations were sampled around Praslin and Mahe islands in January 2019, representing 4 recovering coral reefs (RCR), and 3 macroalgal dominated (mainly *Sargassum* and green turf algae) reefs (MSR) (Figure 1).

The recovering coral reefs had recovered their live coral following the 1998 coral bleaching event [6], but experienced 70% mortality in 2016, having a mean coral cover of 6% by 2017 [40]. Reef ecosystems of the Inner Seychelles support ecologically and phylogenetically diverse fish families. Species in the families *Siganidae*, *Lethrinidae*, *Lutjanidae*, *Acanthuridae*, *Scarinae*, *Mullidae*, *Labridae,* and *Haemulidae* together comprise >95% of total trap fishery catches [42]. Fish samples were collected using handlines and traps deployed from a small boat, using diverse baits (coconut, mackerel, seaweed). In order to take into account intraspecific variability of the gut bacteriome, up to 11 adult individuals of each species were sampled in each site. Immediately after capture, fishes were killed by cervical dislocation (following the European directive 2010/63/UE) and conserved on ice in coolers for dissection in the laboratory later the same day. The animal study was reviewed and approved by the Seychelles Fishing Authority (Memorandum of Understanding signed the 12 December 2018) and by the Lancaster University FST research Ethics review committee (approval number FST18132). At the laboratory (Seychelles Fishing Authority), fishes were placed in trays, washed with 70% ethanol, and the whole intestinal tract of each fish was extracted using sterile dissection tools following the protocol of Clements et al. (2007) [43] and Miyake et al. (2015) [20]. Briefly, we squeezed out the gut content (taking care to avoid contamination by gut wall cells) into a 2 mL sterile Eppendorf tube by rolling a sterile 1 mL micropipette on the intestinal tract starting from segments posterior to the stomach (spanning the midgut and hindgut) or from the 75% most distal part of the gut for fishes lacking stomachs. Gut contents were immediately flash frozen in liquid nitrogen and stored at −80 °C until ready for DNA extractions. A total of 99 fishes belonging to 36 species covering 19 genera and 9 families were sampled for their gut bacteriome (Appendix A).

### 2.2. Fish Identification and Diet Type Definition

For all fishes, host taxonomic identification was performed using the reference book on reef fishes from the West Indian Sea [44]. Fish diet was described using categories as in Mouillot et al. (2014) [45], where Carnivores are separated into invertivores (MI) which mainly feed on mobile invertebrates (i.e., benthic species such as crustaceans) and piscivores (FC) (i.e., feeding on teleosts or cephalopods). Herbivores are divided into strict herbivores (H) eating fleshy macroalgae with browsing (*Siganidae*) and grazing (*Acanthuridae*) behaviors, and detritivores (HD) with scrappers (*Scarinae*), which bite dead pieces of coral and indirectly scrape away turf algae [46]. Finally, omnivorous fishes (OM) feed on both algae or cyanobacteria and small invertebrates (i.e., zooplankton such as copepods). We used a principal coordinates analysis (PCoA), based on Bray-Curtis dissimilarity, to illustrate their distribution through sampled sites (Appendix A). To assess the sources of variation (i.e., taxonomy and diet) in the Bray-Curtis matrix, we used a PERMANOVA analysis based on 1000 permutations [47] with the function adonis, in the vegan package [48].

### 2.3. DNA Extraction and 16S rDNA Gene Amplification

Total genomic DNA from 200 mg of homogenized intestinal contents and from swabs was extracted using the MagAttract PowerSoil^®^ DNA kit according to the manufacturer instructions (MoBio Laboratories, Inc., Carlsbad, CA, USA) with automated processing and the liquid handling system KingFisher Flex™ (ThermoScientific^®^, Waltam, MA, USA). Nucleic acids were eluted in molecular water (Merck Millipore™, Burlington, MA, USA) and quantified on a NanoDrop 8000 ™ spectrophotometer (ThermoScientific^®^, Wilmington, MA, USA). The V4-V5 region of the 16S rDNA gene was targeted with the universal primers 515F-Y(5′-GTGYCAGCMGCCGCGGTAA-3′) and 926R (5′-CCGYCAATTYMTTTRAGTTT-3′) [49] coupled with platform specific Illumina adaptor sequences on the 5′ ends. Each 25 µL PCR reaction was prepared with 12.5 µL Taq Polymerase Phusion^®^ High-Fidelity PCR Master Mix with GC Buffer (New England Biolabs^®^, Inc., Ipswich, MA, USA), 0.5 µL forward primer (10 µM), 0.5 µL reverse primer (10 µM), 1 µL template DNA, 0.75 µL DMSO, and 9.75 µL molecular water. PCR amplifications involved the following protocol: An initial 98 °C denaturing step for 30 s following by 35 cycles of amplification (10 s denaturation at 98 °C; 1 min at 60 °C annealing; 1.5 min extension at 72 °C), and a final extension of 10 min at 72 °C. Amplification and primer specificity were verified by electrophoresis on a 2.0% agarose gel for confirmation of ~450 bp amplicon size. All samples were amplified in triplicate and equally pooled for a final product of 50 µL. Extraction of blank samples used as DNA extraction controls were also performed. None of them were successfully amplified with the primers used in this study. Each amplicon pool was sequenced using the 2 × 250 bp Miseq chemistry on an Illumina MiSeq sequencing platform at the INRA GeT-PlaGE platform (Toulouse, France).

### 2.4. Sequence Processing

All analyses were carried out with R software 3.6.2 (https://www.r-project.org/, accessed on 9 August 2021) [50] and are available on GitHub: https://github.com/mccheutin/Seychelles.git, accessed on 9 August 2021.

Sequence reads were processed using the DADA2 pipeline (v.1.12.1) in R [51], following the pipeline’s tutorial (https://benjjneb.github.io/dada2/tutorial.html, accessed on 9 August 2021). Briefly, sequences were trimmed and filtered based on read quality profiles (maxN = 0; maxEE = (2, 2); truncQ = 2; and truncLen = (240, 240)), error correct, dereplicated and amplicon sequence variants (ASVs) were inferred [52]. Forward and reverse ASVs were merged and pooled in a count table where chimera were identified and removed. Taxonomy assignment was performed using the SILVA reference database (release 132) [53]. The ASVs count table, their taxonomy, and their sequences were organized in a phyloseq object using the phyloseq package (v.1.28.0) [54], on R. ASVs assigned to the kingdom Eukarya, Archaea, and to chloroplast, were removed before computing any further analysis. Bacterial genera known as potential kit contaminants were also removed from our datasets using the list described in Salter et al. (2014) [55]. Overall, 40 genera corresponding to 12% of the total reads were removed (Appendix A). Our final dataset consisted of 1,042,080 sequences belonging to 5129 ASVs.

### 2.5. Defining the Core Bacteriome of Reef Organisms

As observed in many animal microbiomes [18,56], ASVs may span a range from permanent to transient inhabitants. Closely associated ASVs should be more considered when thinking about holobiont ecology [57] since these core taxa may have evolved in close association with their hosts for a long time period [58,59,60]. Here, core bacteriomes were independently identified by examining the species abundance distribution (SAD), patterns of each ASV, and by partitioning the SAD into core and satellite ASVs [61] for the gut (Appendix A) and for the macroalgae (Appendix A). For this purpose, the index of dispersion for each ASV was calculated as the ratio of the variance to the mean abundance (VMR) multiplied by the occurrence. This index was used to model whether lineages follow a Poisson distribution (i.e., stochastic distribution), falling between the 2.5% and 97.5% confidence interval of the χ2 distribution [62]. Index values less than 1 mean that the ASV is under-dispersed compared to the Poisson distribution, so that it spreads uniformly and can be considered as a core ASV. Index values higher than 1 mean that the ASV is over-dispersed, i.e., the ASV is clustered and corresponds to a satellite ASV. Fish and macroalgae core bacteriomes consisted of 531,930 sequences (254 ASVs) and 109,550 sequences (310 ASVs), respectively. All analyses detailed below were performed on the core microbiome.

### 2.6. Inference of ASVs Habitat Preference

We used a BLASTn approach on the nr/nt database and the habitat-associated metadata to the closest ASV match to infer the habitat preference of the 254 ASVs constituting the enteric core bacteriome of reef fishes [22,28]. Only blast results with an identity >95% and a sequence coverage >95% were kept. Information concerning the isolation source contained in the GenBank fields “isolation source”, “host”, and “title” of each closest blast were extracted using a dedicated python script and parsed into “Animal”, “Environment” (i.e., free living bacteria associated with sediment, soil or water), and “Other” habitat categories. For ASVs associated with animals, we further categorized the isolation sources into specific hosts (i.e., fish, marine invertebrates, terrestrial vertebrates, and unknown animals) and organ (i.e., gut, tissue, and other organs) categories (Supplementary File 2). In order to associate these habitat preferences to the phylogenetic affiliation of each ASV, core bacteriome ASVs were aligned against the silva.nr_v132 reference database using mothur v.1.35.1 ([63]; https://mothur.org/, accessed on 9 August 2021) before being imported into the ARB software ([64]; http://www.arb-home.de/, accessed on 9 August 2021) and loaded with the SILVA (v.138) reference database [53]. A base frequency filter was applied to exclude highly variable positions before adding sequences to the maximum parsimony backbone tree using the parsimony quick add marked tool implemented in ARB. The tree and the associated categories were drawn and visualized using the interactive Tree of Life (iTOL) web server ([65]; https://itol.embl.de/, accessed on 9 August 2021).

### 2.7. Computation of Alpha and Beta-diversity of Bacteriomes

In order to correct for the uneven sequencing depth among samples, 1041 sequences were randomly sub-sampled within each sample using the “rarefy_even_depth” function from the phyloseq R-package v.1.28 [54] (Appendix A). Good’s coverage estimator [66] was 99.9 ± 0.1 indicating that the coverage was still excellent after rarefaction. Taxonomic diversity of each microbial community (fish gut or macroalgae swab) was measured using the richness (number of ASV) and the Shannon’s index H, computed on ASV relative abundance, and later exponentially transformed to express it as effective number of species (ENS) [67]. Taxonomic dissimilarities between pairs of bacteriome samples were assessed using the Bray-Curtis dissimilarity computed on relative abundances of ASV.

### 2.8. Functional Diversity Predictions of Bacteriome

Using the 16S rRNA gene information, predictions of metabolic functions for Bacteria were performed using Tax4Fun2 v.1.1.5 [68] with a clustering threshold set at 99%, following the tutorial of the algorithm (https://github.com/bwemheu/Tax4Fun2, accessed on 9 August 2021). In order to account for all ASVs, the predicting functional profiles were then proceeded using the minimum blast identity to reference at 78%. Among the 7279 KOs predicted by Tax4Fun2, about 23% are involved in at least two different metabolic pathways (until 15 for some KOs) and 33% are unknown or hypothetical proteins. These KOs are thus not indicators of a particular function and are a source of an additional and false functional redundancy, hardly ever taken into account in the literature. To avoid this bias, we created a new functional table containing 3261 unique KOs, involved in only one metabolic pathway.

### 2.9. Statistical Tests

First, the gut and macroalgal bacteriomes were compared in richness and in composition by measuring the alpha and beta-diversity. In the same way, to understand the influence of the reef condition on the bacteriomes, we compared the same measures between reef conditions for both macroalgae and fishes. Since the effect of reef condition could have been masked by the effect of diet or phylogeny, we removed this by analyzing the dataset at different community levels (i.e., inside trophic guilds, family and species level). Only levels with at least a triplicate per reef condition were tested. Comparison of alpha diversity indices (richness and entropy) was achieved using a Kruskal-Wallis test (999 permutations) in the vegan R-package followed by a post-hoc Dunn test (999 perm, *p*-value corrected by Bonferroni’s method) in order to identify which group means differed. To determine beta-diversity changes, significant sources of variation in bacteriome Bray-Curtis dissimilarity matrices were assessed using permutational analysis of variance (PERMANOVA) with the adonis function from the vegan package.

ASV biomarkers of bacteriomes of macroalgae, carnivorous, and herbivorous fish were identified using the LEfSe algorithm [69]. The first analysis step was a non-parametric Kruskal-Wallis (KW) sum-rank test allowing the detection of taxa with significant differential abundance. Biological consistency was subsequently investigated using a pairwise Wilcoxon test. Finally, linear discriminant analysis (LDA) was used to estimate the effect size of each differentially abundant taxon. Alpha values of 0.05 were used for KW and Wilcoxon tests and a threshold of 3 was used for logarithmic LDA scores. The same analysis was used to identify functional biomarkers (i.e., KO) of the *Scarinae* and *Siganidae* bacteriomes.

## 3. Results

### 3.1. Sampling Size and Composition of Fish Catch between Reef Conditions

The fish species distribution and sample size were highly variable among reefs. The sampling size of caught fish species was higher in recovering coral reefs (RCR), with 27 species sampled compared to 17 species in macroalgae shifted reefs (MSR). Fish community composition and fish diet behavior differed significantly between RCR and MSR (Appendix A) with a higher abundance of the scrapers *Scarinae* in RCR while grazers *Siganidae* are more abundant in MSR (Appendix A), conforming with the underwater visual census (UVC) data [42]. In contrast, carnivorous species, overall represented by the *Lethrinidae* and *Lutjanidae* families, were distributed in both reefs with 11 *Lethrinidae* in RCR and 15 in MSR and six *Lutjanidae* in RCR and eight in MSR. Only four species have been sampled in triplicate in both RCR and MSR (i.e., *Scarus ghobban*, *Lethrinus mahsena*, *Lethrinus enigmaticus,* and *Aprion virescens*) (Appendix A).

### 3.2. Composition and Diversity of the Fish Core Gut Bacteriome

A total of 254 bacterial ASVs representing 63% of the total reads formed the core bacteriome of the 99 fish gut samples (Appendix A). This core bacteriome was dominated by the *Proteobacteria* (dominated by the order *Vibrionales*) and the *Firmicutes* phyla (mainly constituted by the order *Clostridiales*) that represented collectively more than 67% of the sequences (Appendix A). Other less abundant phyla such as the *Bacteroidetes* (8%), *Fusobacteria* (8%), *Spirochaetes* (5%), *Planctomycetes* (3%), *Cyanobacteria* (3%), *Verrucomicrobia* (3%), and *Tenericutes* (2%) constituted the rest of the fish core bacteriome.

BLASTn analysis revealed that 70% (178) of the bacterial ASVs were closely related to sequences previously retrieved from animal microbiomes (Figure 2). In addition, 45% (115 ASV) belonged to the *Akkermansiaceae*, *Desulfovibrionaceae*, *Vibrionaceae*, *Rikenellaceae*, *Fusobacteriaceae*, and *Lachnospiraceae* families, and matched preferentially sequences previously reported in the intestinal tract of fish from the *Siganidae*, *Acanthuridae*, and *Scarinae* families (Figure 2, Appendix A), indicating a certain degree of conservation for a significant part of the coral fish gut bacteriome.

In addition to these fish gut specialists, 17% (44 ASV) of the core ASVs, mainly affiliated to the *Vibrionaceae*, *Pirellulaceae, Lachnospiraceae,* and *Endozoicomonadaceae* families, were best related to sequences associated with other marine animal bacteriomes, such as corals or sponges, indicating that another significant part of the fish gut bacteriome maybe symbiotic generalists distributed among other marine organisms. The composition of fish core gut bacteriomes differed significantly (PERMANOVA *p* = 0.001; R2 = 0.07) from macroalgae bacteriomes (Appendix A) which were dominated by bacteria from the *Proteobacteria* (56%), *Bacteroidetes* (25%), and *Cyanobacteria* (10%) phyla. Both richness and diversity of fish core gut bacteriomes were half that of macroalgae bacteriomes (Appendix A). Herbivore bacteriome shared 2.5 times more ASVs with the macroalgal bacteriome than the carnivore one (21 vs. 8) (Appendix A), mainly belonging to the Orders *Bacteroidales* (i.e., *Rickenella*), and *Clostridiales* (i.e., *Lachnoclostridium*). Fish gut bacteriomes were also more variable in their composition than macroalgae bacteriomes as indicated by a significantly higher dispersion (Appendix A).

### 3.3. Alteration of the Coral Reef Significantly Disrupts Herbivore but Not Carnivore Bacteriomes

The reef condition explained a small but significant amount of the variability in bacteriome community composition among all fishes (Table 1; Figure 3A).

However, the reef condition neither appeared as a significant driver of variability, nor of bacteriome diversity between fish individuals (Figure 3B). Similarly, the ordination of macroalgae bacteriomes in a PCoA showed a clear separation between CCR and RCR (Figure 3C) which explained 7% of the variance in the community composition for macroalgae bacteriomes (Table 1). In addition, richness of macroalgae bacteriomes were 80% higher in MSR (Figure 3D). For fishes, diet was one of the main drivers of gut bacteriome composition as indicated by a PERMANOVA analysis (Table 1, Figure 3A). The gut bacteriome of herbivores (i.e., grazers, scrapers, browsers, and the two omnivorous *Cantherines pardalis*) was characterized by the enrichment of 12 biomarkers, genera belonging mainly to the *Desulfovibrionales*, *Bacteroidales,* and *Fusobacteriales*, while eight genera belonging mainly to the *Clostridiales* and *Vibrionales* appeared as biomarkers for carnivores (i.e., invertivores and piscivores) (Appendix A).

The reef condition significantly affected the composition of gut bacteriome of herbivorous fishes (R2 = 0.07; *p* = 0.001, Figure 4, Table 1).

However, part of this effect may be driven by the strong fish species turnover among herbivores between coral and macroalgae dominated reefs. Indeed, 16 out of 17 of *Siganidae* and *Acanthuridae* fishes were distributed in MSR, while 19 out of 22 *Scarinae* fishes (mainly represented by *Scarus ghobban*) were present in RCR (Figure 4A). Fish phylogeny was a strong and significant determinant of bacteriome composition at the family (R2 = 0.14; *p* = 0.001), genus (R2 = 0.24; *p* = 0.001), and species (R2 = 0.43; *p* = 0.001) levels. In order to exclude this effect, we analyzed the herbivore dataset at the family and species levels (for *Scarinae* and *Scarus ghobban*, the only herbivores distributed in both reef conditions). In this way, we corroborated the fact that gut bacteriome composition did differ as a result of reef condition (Figure 4B,C). This effect was marginal at the species level probably due to the low number of samples (R2 = 0.28; *p* = 0.078, N = 7). Differences in the composition of herbivore core bacteriomes among reef conditions was driven by changes in the relative abundance of biomarkers of the *Scarinae* and *Siganidae* families (Figure 5).

Bacteriome abundance in MSR was lower for six of the seven *Scarinae* biomarkers and one (i.e., *Anaeroplasma*) was totally absent. *Fusobacterium* and *Odoribacter* biomarkers were only present in *Siganidae* in MSR (Figure 5). These biomarkers accounted for 0.8% on average of the *Scarinae* bacteriomes and 0.4% of the whole dataset. The decrease in *Scarinae* biomarkers was paralleled by a significant decrease in the abundance of 207 specific KOs (Kegg Orthologs) mainly involved in host lipid (i.e., fatty acids, butanoate, propanoate, and glycerophospholipid metabolisms) and glucose homeostasis (Figure 6).

In contrast, *Siganidae* biomarkers, all efficient anaerobes fermenters of plant and algal polysaccharides [30], showed a significant increase in MSR and one new appeared (i.e., *Brachyspira*), accounting for 5.9% on average of the *Siganidae* bacteriomes and 2.0% of the whole dataset. This increase came also with an increase in KOs notably involved in carbohydrates metabolism (starch, sucrose, fructose, and mannose), DNA replication, and nitrogen metabolism suggesting higher rates of fermentation and a stimulation of bacterial growth (Figure 6). Reef condition neither appeared as a significant driver of herbivore bacteriome variability, nor of bacteriome diversity (Appendix A).

Contrary to herbivores, we did not detect a significant effect of reef condition on any of the bacteriome diversity facets (i.e., alpha diversity, beta diversity, and variability) of carnivorous fishes (Table 1, Figure 4, Appendix A). The fish family was the only driver of difference in microbiome composition (R2 = 0.10; *p* = 0.007).

## 4. Discussion

Macroalgal shifted reefs (MSR) are often considered degraded systems in which drastic changes to biotic communities occur, particularly reef fishes [70,71]. So far, the “microbial phase shift” [72] consecutive to a macroalgae regime shift has been studied only in free living microbial communities [73,74,75,76] and primary producer microbiomes [11,77,78]. Here, we pinpointed for the first time the influence of such a shift on the gut bacteriomes of Seychelles reef fishes.

The observations from Robinson et al. (2019) [42] indicated that biodiversity losses were more severe in shifted-reefs resulting in novel fish compositions. This conformed with the different fish functions (e.g., browsing and grazing activities) found in MSR compared to recovering coral reefs (RCR) (Appendix A). Alterations to habitat directly affect coral-dependent fish species [79] such as coral dwellers [80,81] and corallivores [82], and promote the replacement of these highly specialized species by opportunistic species that live in areas of low relief and rubble [70,83,84,85]. In agreement, fish communities from MSR were characterized by a depletion in *Scarinae*, which are scavengers feeding the epilithic layers present on corals [46], and the dominance of browsers and grazers of the *Siganidae* and *Acanthuridae* families [20] (Appendix A). By conditioning the availability of their nutritional resources, regime shifts influenced the occurrence of these two herbivorous fish families (Appendix A). Among opportunistic species, invertivorous fishes are believed to benefit from a carbon flow cascade in which the important release of dissolved organic material in algae-dominated reefs stimulates microbial production ultimately fueling benthic invertebrate biomass [86,87]. In this study, invertivores, essentially represented by fishes from the *Lutjanidae* and *Lethrinidae* families, were however uniformly distributed among RCR and MSR.

Several lines of evidence indicate that microorganisms play an active role in the transition from coral dominance to fleshy algae through the DDAM positive feedback loop (dissolved organic carbon, disease, algae, microorganism) [88,89]. In this mechanism, exudation of labile organic matter by turf and macroalgae promotes an increase in microbial abundance and activity, as well as a change in the composition towards copiotrophic and potentially pathogenic microbial taxa, ultimately causing a physiological deterioration of the coral tissues [11] and a dysbiosis in their microbiome [90]. Except for a recent study [91], disruption of the planktonic microbial composition [73,75,76,92,93] and coral microbiomes [11,12,77,78,90] is a recurrent pattern in MSR. Accordingly, we observed here a significant difference in the composition of macroalgae bacteriomes between MSR and RCR accompanied with an increase in bacterial richness in MSR (Figure 3C,D). Macroalgae bacteriomes in MSR were enriched in *Alphaproteobacteria* (*Ahrensia* sp. and *Albimonas* sp.) and *Gammaproteobacteria* (*Leucothrix* sp.). Enrichment in *Gammaproteobacteria* and particularly from the *Leucothrix* genera, which contains filamentous species known to provoke massive invertebrate egg and larvae mortalities [94], agrees with the DDAM model predicting that a proliferation of macroalgae leads to an increase in copiotrophic microorganisms with the potential to create disease. Altogether, these results indicate a microbialization [75] of the MSR studied here, although we did not assess microbial abundance in our sampling.

We showed that 45% of the ASVs composing the core fish gut bacteriomes corresponded to fish specialists, mainly belonging to the *Desulfovibrionaceae*, *Vibrionaceae*, *Akkermansiaceae*, *Fusobacteriaceae,* and *Lachnospiraceae* families, often retrieved in studies investigating the gut microbiome of coral fishes [21,22,28,30,95,96]. In addition, a significant part of core ASVs were symbiotic generalists shared among marine organisms indicating a potentially important connectivity of fish gut bacteriomes with their surrounding habitat and animal-associated microbial communities, through feeding activity and defecation. This suggests that perturbations of their habitat microbiome related to macroalgal regime shifts could translocate to their own microbiome. Indeed, although fish diet and taxonomy were major determinants of fish gut bacteriome composition, this latter differed significantly between RCR and MSR (Table 1, Figure 4). Shifts in the fish gut microbiome may reflect changes in diet in degraded habitats. While this has never been observed before in coral reef ecosystems, in disturbed continental areas where their nutritional resources were modified or even absent, the composition of black howler monkey enteric microbiomes responded to habitat perturbations [32,97]. Since macroalgae regime shifts represent an important modification of their main nutritional resources, we hypothesized a strong effect on herbivorous fish gut bacteriomes. In agreement, the reef condition explained a significant amount of the variance for herbivorous fish, while we failed to detect any significant effect for carnivorous fishes. One explanation may be related to the fact that carnivorous fishes seem to have a larger dietary niche width than obligate herbivores [98] that would allow them to forage in adjacent healthy areas of the reef [99]. Our sampling strategy did not allow us to detect a significant effect of coral-macroalgal shift at the intra-species level. To overcome this limitation related to the high intra-specific variability observed in fish gut bacteriomes, future investigations should focus on species present in both MSR and CCR and with a significant increase in the number of individuals (more than 10) in each condition and species.

Rather than a dysbiosis, the significant response of herbivorous gut bacteriome composition to the condition of the reef reflected the loss or gain of specific bacterial taxa associated with the strong turnover of their hosts, particularly *Scarinae* and *Siganidae* fishes, between RCR and MSR (Figure 6). This result indicates a certain degree of conservation for a significant part of the coral reef fish gut bacteriome, but also agrees with a recent study showing that loss of the most vulnerable reef animals, and among them fishes, due to reef degradation would induce a significant loss of the reef prokaryotic richness [19]. While we did not observe an erosion of bacteriome diversity in MSR, nor an increase of bacteriome variability among individuals expected under the Anna Karenina principle [100], we did record a significant reduction or loss of *Scarinae* biomarkers and associated functional genes towards the prevalence of bacterial fermenters associated with *Siganidae*. In addition, we also observed a concomitant increase in abundance of KOs involved in carbohydrate metabolism (starch, sucrose, fructose, and mannose), DNA replication, and nitrogen metabolism, suggesting higher rates of fermentation and a stimulation of bacterial growth in MSR. Seaweeds such as *Sargassum* and turf algae are rich in sulfated polysaccharides and high carbohydrate food is well known to promote rates of gastrointestinal fermentation [101]. ASVs constituting *Siganidae* biomarkers were closely related to sequences previously retrieved from *Siganus canaliculatus* (Appendix A). Indeed, bacteria from the genera *Desulfovibrio* (sulfate reducing bacteria), *Rickenella*, *Brachyspira* (anaerobic fermentative bacteria), and *Akkermansia* (mucin degrading bacteria) were found to be part of the core bacteriome of *Siganidae* [102,103], accounting for 5.9% on average of their bacteriomes and 2.0% of the whole dataset. These taxa may be of importance for host digestive function in MSR, in particular for the fermentation of sulfated algal polysaccharides. For example, members of the *Rikenella* genus are known to degrade celluloses into short chain fatty acids (SCFA) available for the host through microbial fermentation [30,104]. The prevalence of these fermentative bacteria is in line with the high fermentation rates observed within herbivorous fish hindguts [105], particularly in *Siganidae* [106] and further suggest a well-suited adaptation of *Siganidae* bacteriomes to the consumption of algae. We acknowledge that these predicted functions based on barcoding data should be corroborated by future transcriptional or proteomic studies that could address the consequence of coral-macroalgal shift on the fermentative activity of microbes associated with reef fish. Nonetheless, the predominance of fermentative metabolisms in MSR indicated that regime shifts not only affect the taxonomic composition of fish bacteriomes, but has the potential to also affect ecosystem functioning through microbial functions.

## 5. Conclusions

Identifying the mechanisms and consequences of bleaching-induced benthic regime shifts for reef microbiota is vital for understanding the resilience of these habitats to changing ocean conditions. Here, we showed that a “microbial phase shift” occurred following a macroalgae regime shift, which was translocated to the gut bacteriome of herbivore reef fishes affecting their composition and potentially their functional role in the reef ecosystem. This response reflected the loss or gain of specific bacterial taxa associated with the strong turnover of their hosts between RCR and MSR. A pattern that maybe reflects a long-term effect of regime shifts. The consequences of increasing recurrence of “coral-macroalgae regime shifts” on reef animal microbiota and reef functioning is an emerging field of reef ecology. Further work should investigate the repercussions of microbiota dysbiosis consecutive to habitat degradation impacts on both host fitness and ecosystem functioning.

## Figures and Tables

**Figure 1 microorganisms-09-01711-f001:**
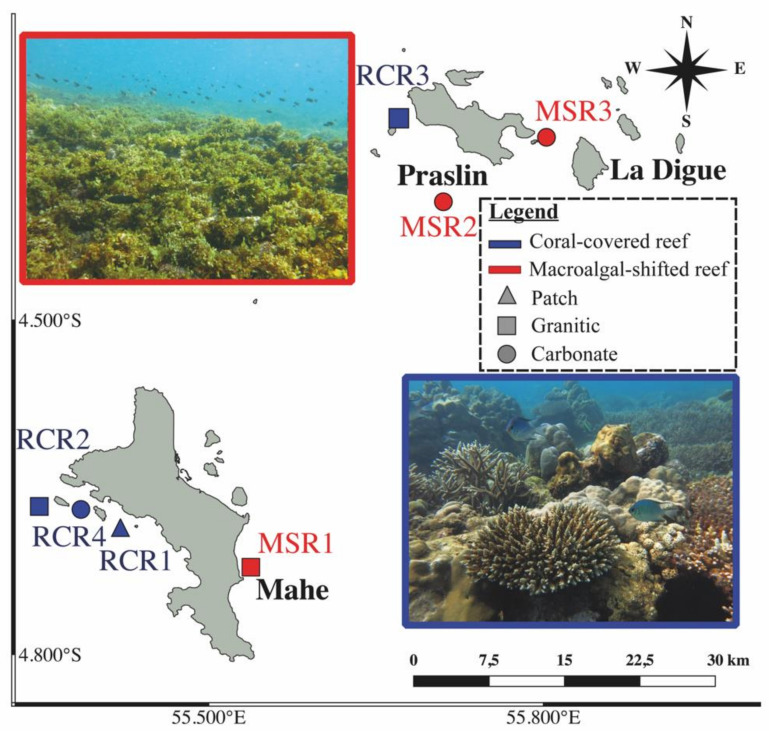
Sampling map of coral-dominated reefs (RCR1: RCR4 in blue) and macroalgae shifted reefs (MSR1: MSR3 in red) with their respective geomorphology (patch, carbonate or granitic). The pictures of coral-covered and macroalgae shifted reefs are represented respectively in blue squared (bottom right) and in red squared (top left). Photo credits: Nicholas A.J. Graham.

**Figure 2 microorganisms-09-01711-f002:**
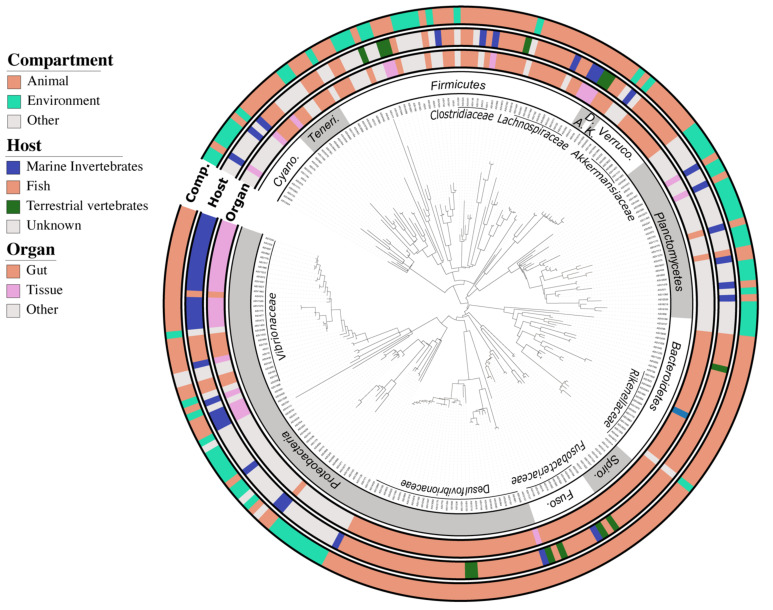
Maximum parsimony phylogenetic tree of the 254 ASVs from the fish gut core bacteriome. The 16S rRNA sequences were inserted into the original SILVA (release 138.1) tree using parsimony criteria with the Bacteria filter excluding highly variable positions. The inner ring represents the order level nomenclature following the taxonomy provided by default in the SILVA bacterial tree. The three outer rings depict the habitat preferences of each ASV described here as three categories (i.e., habitat, specific host, and organ) clustered from the environmental information associated with each closest blast. The tree was drawn using the web-based interface interactive tree of life (iTOL). Abbreviations: Cyano. = *Cyanobacteria*; Teneri. = *Tenericutes*; A. = *Actinobacteria*; D. = *Deferribacteres*; K. = *Kiritimatiellaeota*; Verruco. = *Verrucomicrobia*; Spiro. = *Spirochaetes*; Fuso. = *Fusobacteria*.

**Figure 3 microorganisms-09-01711-f003:**
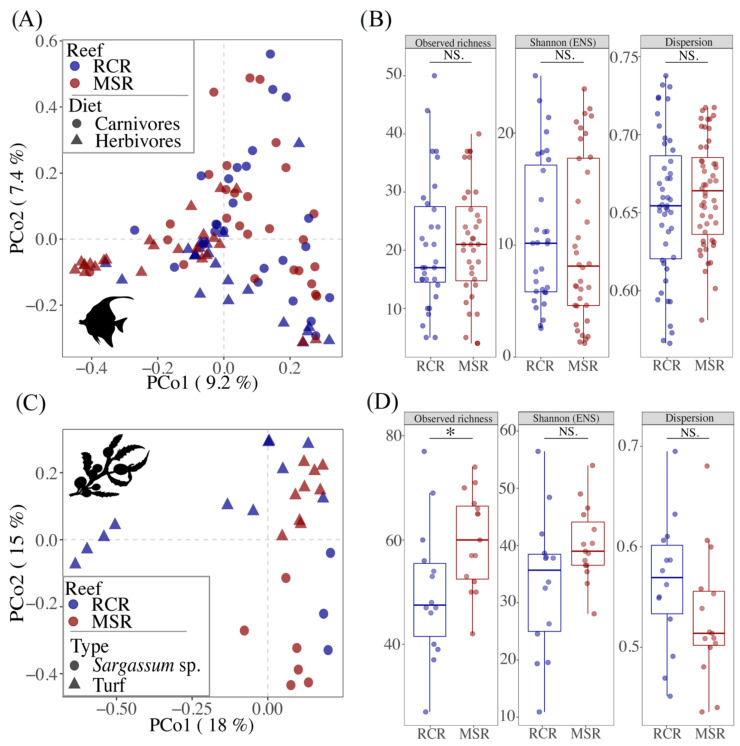
Comparison of the alpha and beta diversity of fish gut and macroalgae bacteriomes in function of the condition of the reef (i.e., coral covered vs. macroalgae shifted). (**A**,**C**) Principal coordinates analysis (PCoA) plots illustrating Bray-Curtis distances between pairs of bacteriome samples. Bacteriomes are colored according to the reef condition, while the shape represents (**A**) fish diet or (**C**) macroalgae type. (**B**,**D**) Boxplots representing the alpha diversity, expressed as the observed richness and the Shannon’s index H-exponentially transformed in effective number of species (ENS), and the dispersion (distance to the centroid for each sample type grouping) calculated for each bacteriome sample. Horizontal brackets indicate pairs which differ significantly: *** ≤ 0.001; ** ≤ 0.01; * ≤ 0.05) or not (NS) with a Wilcoxon test.

**Figure 4 microorganisms-09-01711-f004:**
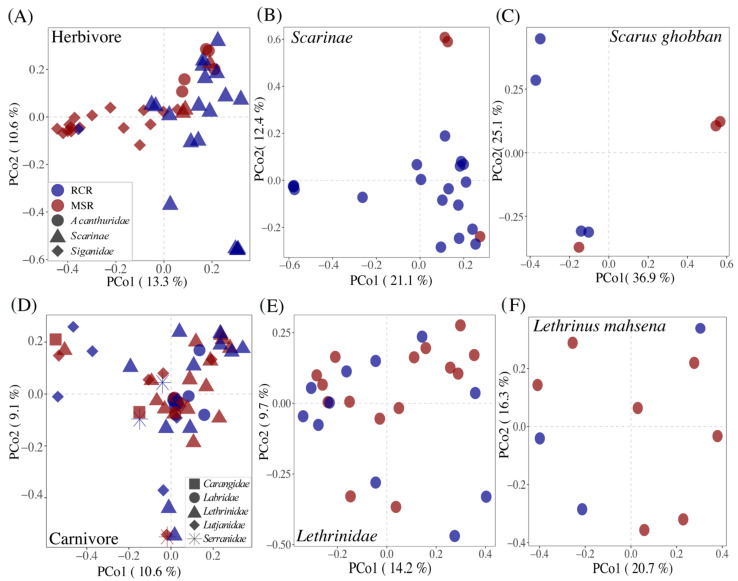
Principal coordinates analysis (PCoA) plots illustrating Bray-Curtis distances between pairs of bacteriome for (**A**) herbivorous fishes, (**B**) *Scarinae*, (**C**) *S.ghobban*, (**D**) carnivorous fishes, (**E**) *Lethrinidae,* and (**F**) *L. mahsena*. Bacteriomes are colored according to the reef condition: RCR in blue and MSR in red. See Appendix A for the *Lutjanidae*, *A. virescens,* and *L. enigmaticus* results.

**Figure 5 microorganisms-09-01711-f005:**
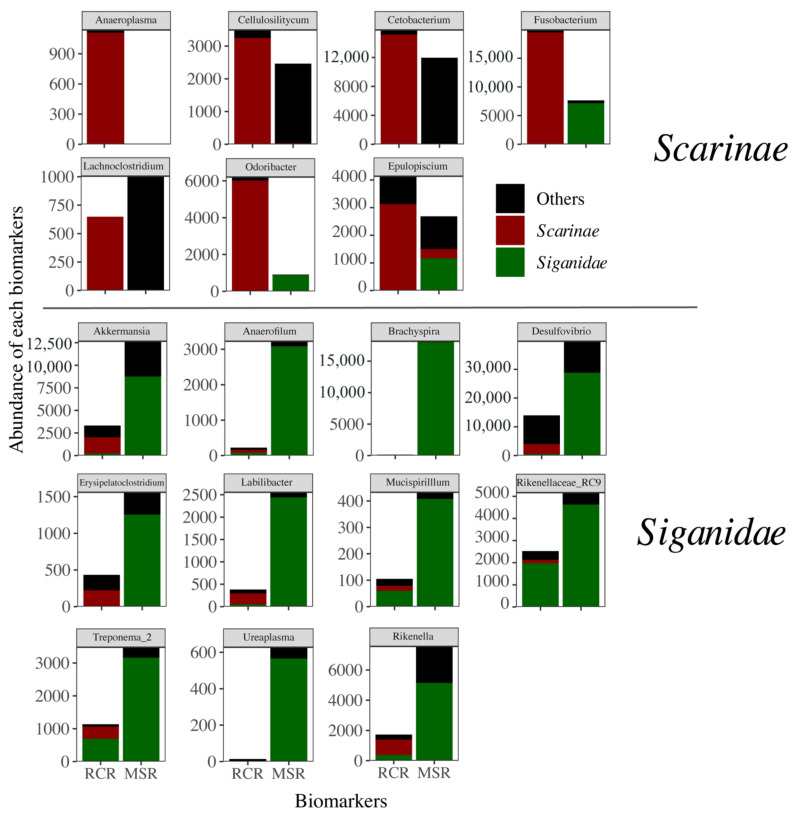
Abundance of each biomarker at Genus level related to *Scarinae* and *Siganidae* (delineated using a LEFSE approach) in RCR and MSR. Contribution of *Scarinae* (red), *Siganidae* (green), and other fish families (black) is indicated on each biomarker.

**Figure 6 microorganisms-09-01711-f006:**
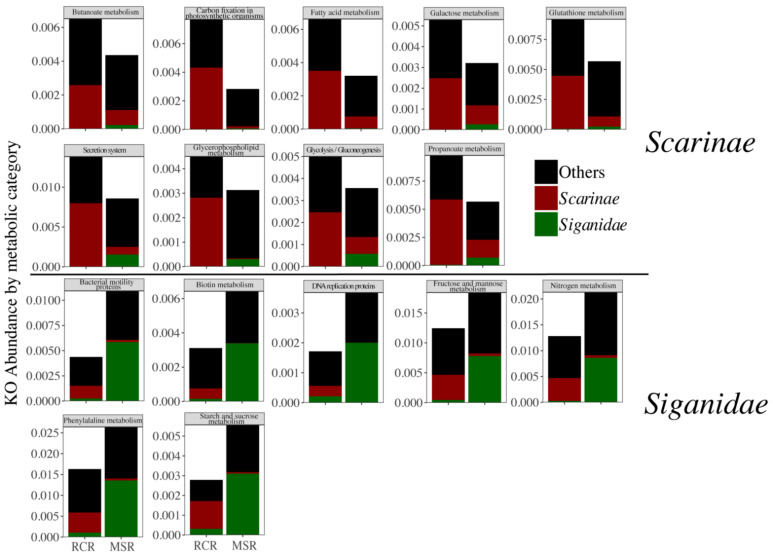
Abundance of each Kegg Ortholog (KO) merged by metabolic pathway, related to enteric bacteriomes of Scarinae (red), Siganidae (green), and other fish families (black) (delineated using a LEfSe approach) in RCR and MSR.

**Table 1 microorganisms-09-01711-t001:** Results of PERMANOVA on the 29 core bacteriomes of macroalgae and 99 enteric core bacteriomes of reef fish (*n* = sampling size). For a relevant sampling size (If not “–“), diet, taxonomy and the reef condition (RCR vs. MSR) were tested (999 perms). Signif. codes for *p*-value: *** ≤ 0.001; ** ≤ 0.01; * ≤ 0.05 or not (NS).

	Diet	Family	Genus	Species	Reef condition
Algae (*n* = 29)	_	_	_	**R2 = 0.13** **(***)**	**R2 = 0.07 ** **(**)**
Fish (*n* = 99)	**R2 = 0.04** **(***)**	**R2 = 0.16** **(***)**	**R2 = 0.27** **(***)**	**R2 = 0.46** **(***)**	**R2 = 0.02** **(***)**
Herbivores (*n* = 44)	_	**R2 = 0.14** **(***)**	**R2 = 0.24** **(***)**	**R2 = 0.43** **(***)**	**R2 = 0.07** **(***)**
*Scarinae* (*n* = 22)	_	_	**R2 = 0.14** **(**)**	_	**R2 = 0.09** **(**)**
*S.ghobban* (*n* = 7)	_	_	_	_	R2 = 0.28 (NS.)
*Siganidae* (*n* = 17)	_	_	_	**R2 = 0.20** **(**)**	_
Carnivores (*n* = 53)	_	**R2 = 0.10** **(**)**	**R2 = 0.24** **(**)**	**R2 = 0.44** **(*)**	R2 = 0.02 (NS.)
*Lutjanidae* (*n* = 14)	_	_	R2 = 0.08 (NS.)	R2 = 0.34 (NS.)	R2 = 0.10 (NS.)
*A.virescens* (*n* = 7)	_	_	_	_	R2 = 0.17 (NS.)
*Lethrinidae* (*n* = 26)	_	_	_	R2 = 0.21 (NS.)	R2 = 0.05 (NS.)
*L.mahsena* (*n* = 10)	_	_	_	_	R2 = 0.12 (NS.)
*L.enigmaticus* (*n* = 6)	_	_	_	_	R2 = 0.17 (NS.)

## Data Availability

Raw Illumina Miseq sequence data for each sample obtained in this study were deposited in the NCBI Sequence Read Archive (SRA) under BioProject accession no. PRJNA674042.

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
