# Peer review of "Microbial Shift in the Enteric Bacteriome of Coral Reef Fish Following Climate-Driven Regime Shifts"

_microorganisms, 2021, doi:10.3390/microorganisms9081711_

Round 1
Reviewer 1 Report
Marie-Charlotte Cheutin and colleagues here present the manuscript entitled ‘microbial shift in the enteric bacteriome of coral reef fish following climate-driven regime shifts’. The authors use 16S rRNA metabarcoding together with functional attribution analysis (i.e., inference of putative functions) to compare fish gut microbiomes between recovered (coral-dominated) and degraded (macroalgae dominated) reefs in the Seychelles. Gut microbiome compositions differed significantly between recovered and degraded sites, which is also reflected in inferred metabolic potential of associated gut bacteria.
This is a well-executed study and an exceptionally well-written paper that tells a compelling story. I have no doubt that it will be of interest to a broad audience. I have no major concerns.
The only minor point I would like the authors to look into, are the Propionibacteria mentioned in line 297. These are human skin bacteria, and their presence in the dataset may be due to contamination (yes they do also crop up in coral microbiome datasets, but as frankly coral microbiome researchers are notorious for overlooking contaminants in 16S data). In contrast to contaminants present in the DNA extraction kit (which have been considered by the authors), contamination from handling samples may not be as consistent across the sample set, but may crop up in individual samples only (depending on the level of contamination). Notably, Propionibacteria are also listed in Salter et al. (2014). As long as the proportion of ASVs affiliated to Propionibacterium is not driving patterns in the authors’ data set, everything should be fine, but I advise the authors double check that.
Author Response
Dear Reviewer 1,
Thank you for the comments. Please, see the attachment for the responses.

Reviewer 2 Report
This article has scientific quality and presents some novelty. Nevertheless, some minor considerations have to be clarified or fixed.
Title: Please remove the dot from the title. With the title, the story you want to pass on to the scientific community just begins, so it is not wise to put a kind of conclusion as a title. Please the authors consider to revise it.
Line 90 -95 & 393-395. Where is the “goal” of the present study? It's not clear the main message of the study.
Section «Fish identification and diet type definition»; Line 130-131 «Fish diet was described using categories as 130 in Mouillot et al., (2014)[45]». Please indicate in more detail the diet type definition. Also mention the results in the appropriate section.
Please discuss the methodological limitations of the article.
Author Response
Dear Reviewer 2,
Thank you for the comments. Please see the attachment for the responses.
